# Elucidating the Correlation between Bone Mineral Density and Multifidus Muscle Characteristics: A Cross-Modal Study with Dual-Energy X-ray Absorptiometry and Spinal Computed Tomography Texture Analysis

**DOI:** 10.3390/diagnostics13223466

**Published:** 2023-11-17

**Authors:** Min-Woo Kim, Young-Min Noh, Yun-Sung Jung, Se-Yeong Jeon, Dong-Ha Lee

**Affiliations:** 1Department of Orthopedic Surgery, Busan Medical Center, 62, Yangjeong-ro, Busanjin-gu, Busan 47227, Republic of Korea; drkimminwoo@naver.com (M.-W.K.); doctornoh77@naver.com (Y.-M.N.); yunpang9@gmail.com (Y.-S.J.); 2Ulsan National Institute of Science and Technology, Ulsan 44919, Republic of Korea; skiende74@gmail.com

**Keywords:** dual-energy X-ray absorptiometry (DXA), computed tomography Hounsfield unit (CT HU), bone mineral density (BMD), multifidus muscle, morphometric texture analysis

## Abstract

Background: Recent research underscores the clinical relevance of muscle conditions such as sarcopenia and their links to bone mineral density (BMD), yet notable gaps persist in the understanding of their interconnections. Our study addresses this by introducing a novel approach to decipher the correlation between BMD and the texture of the multifidus muscle, utilizing spinal computed tomography (CT) and dual-energy X-ray absorptiometry (DXA) to evaluate muscle texture, BMD, and bone mineral content (BMC) at the total lumbar vertebra and total hip. Methods: Our single-institution study examined 395 cases collected from 6 May 2012 to 30 November 2021. Each patient underwent a spinal CT scan and a DXA scan within a one-month interval. BMD and BMC at the total lumbar vertebra and total hip were measured. The texture features of the multifidus muscle from the axial cuts of T12 to S1 vertebrae were assessed via gray-level co-occurrence matrices. CT texture analysis values at angles of 45 + 45 and 90 degrees were calculated and correlated with BMD and BMC. A regression model was then constructed to predict BMD values, and the precision of these correlations was evaluated using mean square error (MSE) analysis. Results: Total lumbar BMC showed a correlation of 0.583–0.721 (MSE 1.568–1.842) and lumbar BMD of 0.632–0.756 (MSE 0.068–0.097). Total hip BMC had a correlation of 0.556–0.690 (MSE 0.448–0.495), while hip BMD ranged from 0.585 to 0.746 (MSE 0.072–0.092). Conclusions: The analysis of spinal CT texture alongside BMD and BMC measures provides a new approach to understanding the relationship between bone and muscle health. The strong correlations expected from our research affirm the importance of integrating bone and muscle measures in the prevention, diagnosis, and management of conditions such as sarcopenia and osteoporosis.

## 1. Introduction

Ageing invariably brings about noticeable changes, such as a decrease in bone mineral density (BMD), muscle mass, and strength. These shifts escalate the risk of disability, falls, fractures, and frailty in the elderly, posing a substantial clinical and public health concern. A myriad of shared factors, encompassing genetic, nutritional, lifestyle, and hormonal influences, have been attributed to muscle and bone health [1,2,3,4,5], with their interactions shaping bone strength. Dynamic muscle loading, consequential to muscle contractions and ground impacts during weight-bearing activities, triggers adjustments in weight-bearing bones [6]. Decoding this symbiotic relationship can steer us towards interventions that can bolster musculoskeletal function and curtail adverse clinical outcomes. 

In the geriatric population, the predisposition towards functional disabilities, fall incidents, bone fractures, and the manifestation of frailty is escalating, presenting an undeniable clinical and societal challenge. Recent scientific investigations have shed light on the intricate nexus between muscle and bone, revealing shared determinants spanning genetics, nutrition, habitual lifestyle, and hormonal interplay [1,3,5]. Furthermore, an interdependent relationship emerges when considering bone strength, where one posits that the dynamic load exerted on weight-bearing bones, originating from muscle contractions and ground impact during locomotion, plays a pivotal role [5]. Delving into this symbiotic relationship could provide a blueprint for tailored interventions, targeting the enhancement of musculoskeletal health and subsequently curtailing adverse outcomes, prominently falls and fractures [6]. The bedrock of evidence underscoring this muscle–bone interplay in an aging demographic predominantly stems from meticulous observational epidemiological research.

Existing research, predominantly drawn from older women, consistently highlights a positive correlation between lean mass and both whole-body and regional areal bone mineral density (BMDa (g/cm^2^)) [2,7,8,9]. Some studies even underscore the role of relative appendicular skeletal muscle mass (RASM) in influencing regional BMDa [10]. Despite the crucial role of lean mass, the influence of fat mass on BMDa is disputed [11,12]. However, the link between muscle strength and BMDa in postmenopausal women, although contingent on lean mass, is well established [9]. Sarcopenia, characterized by low muscle mass, has also been connected to lower BMDa [10,13].

In contrast, data from male subjects suggest a more complex interplay between bone and body composition. Studies indicate both positive [7,14] and negative [15] correlations between fat mass and BMDa, while others focus solely on lean mass [16] or RASM [2,3]. Some even contest the lean mass–BMDa correlation after adjustments for BMI [4] or skeletal size [17]. Similar to the findings in women, muscle strength in men has been recognized as a determinant of BMDa regardless of weight, but not independent of lean mass [9]. A comprehensive examination of the link between BMDa and sarcopenia in men, especially following a rigorous definition of sarcopenia, is notably lacking [18]. 

To fill these gaps, we propose a novel approach, inspired by the trabecular bone score (TBS) [19], but utilizing the gray-level co-occurrence matrix (GLCM) [20,21,22], to extract 45 distinct texture analysis values from CT scans. Harnessing machine learning, we aim to establish correlations and build a predictive model, leveraging our rich dataset and follow-up patient data. The objective is to trace the temporal dynamics of bone health.

Our goal is to provide an objective method to quantify BMD at varying scan intervals and monitor its changes over time using our innovative CT model. We aim to transcend traditional gender-based research, encompassing a wider demographic spectrum, including men, women, the elderly, and the young.

### Objectives

We deploy a cross-modal approach, integrating spinal CT and DXA scans conducted within a one-month period, adhering to the concept of opportunistic CT. While we steer away from conventional measures like RASM, our primary focus rests on the multifidus muscle. Through this approach, we intend to shed light on the intricate relationship between BMD and core muscle, thereby enriching the understanding of musculoskeletal health.

## 2. Materials and Methods

### 2.1. Study Design and Approval

This research, carried out between 9 May 2011 and 30 November 2022, was approved by the institutional review board (P01-202109-21-014). The study’s design revolved around texture analysis’s capability of monitoring bone mineral status, especially with a focus on sarcopenia and the evaluation of the multifidus muscle.

Our initial cohort consisted of 1722 cases from 863 patients who had undergone both spine CT and DXA scans at the same institution. To ensure data relevance, we applied a selection criterion of a temporal gap of less than one month between the CT and DXA scan dates, resulting in a refined cohort of 395 cases from 248 patients. Exclusions were made for cases with an absence of a measurable axial cut between the T12 and S1 vertebrae in the CT images; documented instances of compression or burst fractures between T12 and S1; previous surgical interventions such as vertebroplasty or kyphoplasty for spinal compression fractures; the presence of metal artifacts from unstable burst fractures; and difficulties in discerning trabecular bones due to severe osteolytic or pathological changes. After making these exclusions, our analysis was based on a final cohort of 395 cases from 248 patients, as depicted in Figure 1 (Table 1).

By opting for a stringent selection and focusing on an extended region of interest from T12 to S1, this study aims at using a holistic approach to evaluate sarcopenia, with a special spotlight on the multifidus muscle’s role. This methodical approach ensures a comprehensive and insightful exploration of sarcopenia’s implications for musculoskeletal health.

### 2.2. CT and DXA Imaging Protocols

CT scans were conducted utilizing a Siemens SOMATOM 128, Definition AS+ scanner (Siemens Healthcare, Forchheim, Germany). Each scan followed a uniform protocol, comprising a single-energy CT scan set at 120 kVp and 247 mA, with dose modulation and a collimation of 0.6 mm. An effective pitch of 0.8 was sustained, and a B60 (sharp) reconstruction kernel was applied. The spine CT scans, performed without contrast, maintained a reconstructed slice thickness of 5.0 mm. For DXA, we employed a standardized instrument according to the typical guidelines (GE Lunar Prodigy, GE Healthcare, Seoul, Republic of Korea) [23], with reports generated using specialized software (Physicians Report Writer DX (Version 11.4); Hologic, Discovery, WI, USA). Our rigorous commitment to these standardized protocols ensures consistent and replicable results.

### 2.3. Regions of Interest

For accurate statistical assessments from muscle imagery, our regions of interest (ROIs) focused solely on the section of the multifidus muscle, sidestepping potential measurement biases. Among the diverse techniques for defining ROIs, we opted for the thresholding method [24]. A 2-dimensional (2D) slice image was chosen from the spine CT’s axial cut for every patient, capturing the widest axial representation of the multifidus muscle between T12 and S1. As illustrated in Figure 2, we executed our texture analysis within a hand-drawn region encompassing the majority of the muscle space. This approach facilitated an in-depth evaluation over an extensive ROI, enriching the depth and scope of our insights.

#### Estimation of Multifidus Muscle Texture Analysis Values Using CT

Figure 2 delineates the flow of multifidus muscle texture analysis estimations. Specifically, from the spine CT axial cuts of each patient, we selected a slice image showcasing the maximum axial muscle area of the multifidus muscle. In total, 45 feature values {x_j}_(j = 1, …, 45) were garnered from each area. Out of these, five features were rooted in an intensity histogram (of CT HU values), while the remainder stemmed from the gray-level co-occurrence matrix (GLCM), a well-regarded method in texture analysis. GLCM functions (Table 2) spotlight the texture of an image, fetching statistical metrics from a matrix detailing the frequency of specific pixel pair values in an image [18,19]. As reflected in Table 2, we employed diverse statistics (k) in the histogram and an assortment of directions (l), levels (m), and statistics (n) in GLCM. The feature index was denoted as j = k + n + 5 (m − 1) + 20 (l − 1).

Given that each patient had axial cuts from both the right and left of the multifidus muscle of the spine CT, a combined total of 90 features (45 from each side) were accumulated. We used these features in tandem to calculate a comprehensive correlation value. A MATLAB™ software (R2023b) function was deployed to produce nonsymmetric versions of the matrices.

Two linear regressors were employed to extract information from the CT pertaining to multifidus muscle texture analysis values. The primary regressor utilized 45 features from one case to estimate a value for each scan, while the secondary regressor was designed with 90 combined features to capture a more comprehensive texture analysis. Each regression output is represented as a linear amalgamation of features and one bias: y^_j = ∑_(j = 1)^J▒〖w_j x_j〗 + b, where x_j signifies the jth feature. Optimal parameters were determined through a minimization approach, correlating with the DXA reference.

Moreover, to elevate the transparency of the regression models, LASSO was incorporated. LASSO applies the l_1 penalty for sparsity, thereby refining the model’s interpretability. The model parameters are adjusted based on the regularization strength (λ), with a larger λ yielding a sparser solution {w_j^*}.

## 3. Results

### 3.1. Patient Demographics

Our study encompassed a total of 395 cases from 248 patients, with 115 men and 133 women participating. The average age and BMI of the participants were 63.12 ± 10.16 years and 24.09 ± 4.45 kg/m^2^, respectively. The average time interval between the spinal CT and DXA was 7.13 ± 6.12 days (Table 3).

### 3.2. Correlation Analysis

In our research, the texture analysis values derived from the spinal CT scan axial cuts were compared with the BMD and BMC measurements obtained via DXA. Our analysis revealed nuanced correlation strengths between the texture values obtained from the spinal CT scans and DXA measurements. The total lumbar BMC showcased a moderate to strong correlation, represented by values of 0.583 to 0.721 (MSE 1.568–1.842). In contrast, the total lumbar BMD exhibited a pronounced strong correlation, delineated by figures ranging from 0.632 to 0.756 (MSE 0.068–0.097). Meanwhile, the correlation for total hip BMC leaned more towards the moderate spectrum, with values spanning 0.556 to 0.690 (MSE 0.448–0.495). Lastly, the total hip BMD correlation hovered between moderate and strong, marked by a spectrum of 0.585 to 0.746 (MSE 0.072–0.092).

Figure 3, Figure 4, Figure 5 and Figure 6 vividly represent these findings, showcasing scatter plots that draw a link between the texture values of multifidus muscles and the actual BMD and BMC measurements ascertained via DXA.

It is pivotal to emphasize the scope of our study, which ventured into the realm of spinal CT texture analysis, investigating its capacity to serve as an indirect metric for muscle health, particularly in relation to conditions like osteopenia, osteoporosis, and sarcopenia. While the correlations with DXA measurements were indeed statistically significant, they presented a modest intensity, which might hint at intrinsic aspects tied to DXA scanning and the muscle health assessment potential of CT texture analysis.

Our endeavor is distinctive, marking a preliminary exploration that employs an expansive ROI—from T12 to S1—as opposed to the conventional DXA analysis which typically covers L1 to L4. Even though the correlation might appear reserved, this methodology paves the way for alternative avenues in muscle health evaluation, especially in situations where DXA scans might not be the optimal choice.

## 4. Discussion

Our research focused on the potential of machine learning methodologies, specifically linear regression (LR) models [25], and texture analysis of computed tomography Hounsfield units (CT HUs), in the detection and assessment of muscle and bone health. By extracting a broad set of features from spinal CT scans, we aimed to provide estimates of BMD that correlate with DXA measurements.

In our study, the texture analysis from spinal CT scans was juxtaposed with DXA’s BMD and BMC metrics. The correlations we observed offer meaningful insights. For instance, the total lumbar BMC’s moderate to strong correlation (0.583–0.721; MSE 1.568–1.842) suggests a significant association between the spinal textures and bone mineral content. Moreover, the notably robust correlation for lumbar BMD (0.632–0.756; MSE 0.068–0.097) underscores the potential diagnostic capabilities of spinal CT texture analysis. The moderate link with total hip BMC (0.556–0.690; MSE 0.448–0.495) raises questions on regional variability in muscle health, and the hip BMD’s range (0.585–0.746; MSE 0.072–0.092) bolsters the merit of CT-derived metrics in clinical assessments. These findings accentuate the diagnostic relevance of spinal CT texture, potentially broadening its application beyond conventional contexts.

Furthermore, we adopted a broader range of ROIs from T12 to S1 for CT texture analysis, as opposed to the typical L1 to L4 range used in DXA. This opportunistic usage of the existing spinal CT scans not only leveraged an underutilized imaging resource but also widened the area of investigation, offering a potentially more comprehensive assessment of spinal bone health.

In contrast to previous studies, our investigation is not limited to the L1 region. Instead, we considered a broader range, spanning from T12 to S1, a factor that enhances the robustness of our findings [26,27]. We chose to employ spine CT scans for this texture analysis due to their frequent usage during health checkups, often alongside DXA scans for osteoporosis diagnosis. These scans encompass regions of interest (T12-S1), which are integral to DXA BMD measurements. Moreover, the CT HU measurement provides a straightforward representation of BMD using the tissue density of vertebral trabecular bone mass. To further refine this process, we utilized the gray-level co-occurrence matrix (GLCM), a widespread method in texture analysis. Extracted statistical parameters from GLCM, including energy, contrast, entropy, and others, facilitate a quantitative understanding of the spatial relationship between pixels in the analyzed area [22].

BMD, as valuable as it is, might not capture the entire essence of bone health. Hence, our method, which compared BMD with multifidus muscle’s ROI on CT scans using texture analysis, is meant to complement, not replace, traditional muscle quantity measurements. It brings to light additional aspects of bone and muscle health, particularly the bone’s microarchitecture, which plays a pivotal role in fracture risk assessment [28,29]. 

While we acknowledge the importance of considering various modulating factors such as physical activity, medications and drugs, and supplements intake, our current study primarily aimed to establish a generalized correlation by quantifying the multifidus muscle through DXA and CT texture analysis across a diverse demographic. The objective was to explore a broader understanding, setting a basis for more specialized investigations in the future where these modulating factors can be intricately analyzed to unveil their nuanced influences on muscle and bone health.

Our approach, which melds the traditional DXA BMD and our innovative CT texture analysis, offers a dual perspective on bone health [20]. While DXA BMD continues to be the cornerstone for assessing bone mineral content and density, the texture analysis method offers a deeper dive into the intricate microstructural details of the muscles, giving practitioners a more robust diagnostic tool.

Building on these results, the study shows potential avenues for the application of machine learning in the field of medical imaging. By employing traditional radiomics steps, including pre-processing, manual segmentation, and feature extraction, we were able to predict BMD. This implies that CT HU texture analysis could serve as an effective tool for BMC estimation, offering a promising alternative to conventional DXA imaging.

In this study, we harnessed the power of machine learning, specifically the artificial neural network (ANN) and a straightforward linear regression (LR) model, to enhance the precision and efficiency of our analysis. The rapid advances in computing power have made machine learning a transformative force in various fields, including medical imaging, where it is significantly enhancing diagnostic accuracy.

A subset of machine learning, radiomics, has grown exponentially due to its capacity to extract quantifiable features from regions of interest (ROIs) in images. These features play a crucial role in achieving prognostic or predictive objectives. In this study, we utilized conventional radiomics steps, such as pre-processing, manual segmentation, and feature extraction, to predict bone mineral density (BMD). The features we focused on included energy, kurtosis, and skewness from intensity, as well as texture analysis employing the gray-level co-occurrence matrix (GLCM).

However, our study still carries certain limitations. Firstly, the study sample was solely recruited from one medical unit, which might introduce a geographical bias and could potentially distort the results as the sample may be typical for the analyzed territorial area. Efforts for multi-center collaboration in future research could help in generalizing the findings. Secondly, this study was cross-sectional, preventing us from determining the time sequence of the observed relationships, and prospective data would be essential to overcome this limitation. Thirdly, the sample size is relatively small and lacks diversity as factors such as BMI, age, and sex were not thoroughly considered in the analysis, limiting the external validity of our findings. Particularly, in postmenopausal women, we have taken into account the heightened osteoporotic risk factors that lead to increased bone loss during aging. This consideration allows for a more nuanced and comprehensive understanding of the various factors modulating health and influencing BMD in the study population. Finally, all samples were collected over quite an extended period. This extended timeframe raises the potential for variability and errors in the CT and DXA analyses due to different individuals performing the analyses, possibly affecting the consistency and reliability of the results. Nevertheless, these constraints do not overshadow our primary finding: the value of CT scans for multifidus muscle estimation, particularly considering their potential in screening patients for sarcopenia risk without additional diagnostic tests.

In recognizing the multifactorial influences on health, we acknowledge that comorbidities are pivotal factors that can modulate the outcomes of interest in our study. While our primary focus has been on the quantification of the multifidus muscle, future iterations of this research will aim to incorporate a more comprehensive view by considering the impact of various comorbidities, enabling a more holistic understanding of their contributory roles.

Although our research is a significant stride towards integrating machine learning with radiomics for muscle quality assessment, it also highlights the need for further research in this area. Given the modest to strong correlations achieved, CT-derived multifidus muscle estimates may not yet be ready to serve as an additional diagnostic tool for sarcopenia. However, they do offer an alternative perspective, which, in conjunction with traditional methods like RASM, could potentially provide a more holistic and accurate assessment of bone health.

In summary, our study underscores the potential of CT HU texture analysis and machine learning models in providing a new lens for sarcopenia detection and monitoring. Despite certain limitations, this approach offers a promising starting point for future studies aiming to leverage the underutilized resource of spinal CT scans in muscle and bone health assessment. Further investigations involving larger and more diverse patient groups, and refined methodologies, could potentially corroborate and expand upon our findings.

## 5. Conclusions

The analysis of spinal CT texture alongside BMD and BMC measures provides a new approach to understanding the relationship between bone and muscle health. The strong correlations expected from our research affirm the importance of integrating bone and muscle measures in the prevention, diagnosis, and management of conditions such as sarcopenia and osteoporosis.

## Figures and Tables

**Figure 1 diagnostics-13-03466-f001:**
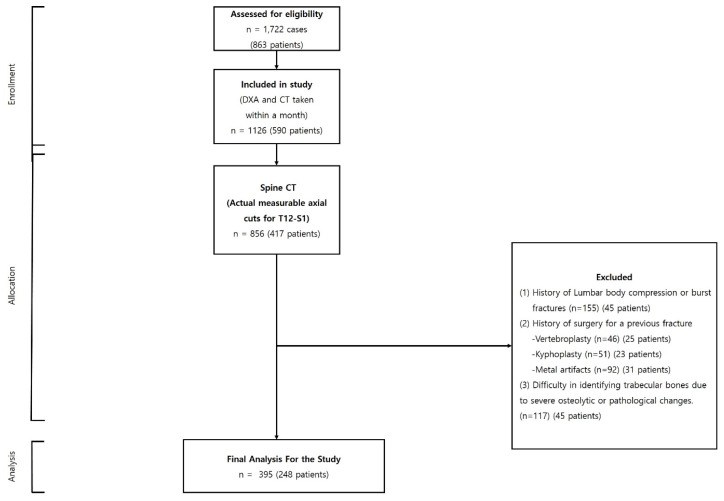
Flowchart illustrating the selection process of patients undergoing concurrent spine CT and DXA scans.

**Figure 2 diagnostics-13-03466-f002:**
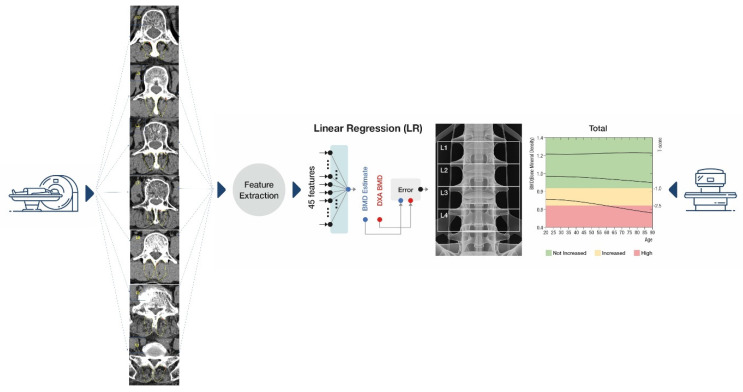
Schematic flow for BMC and BMD estimations from computed tomography. BMC, bone mineral content; BMD, bone mineral density.

**Figure 3 diagnostics-13-03466-f003:**
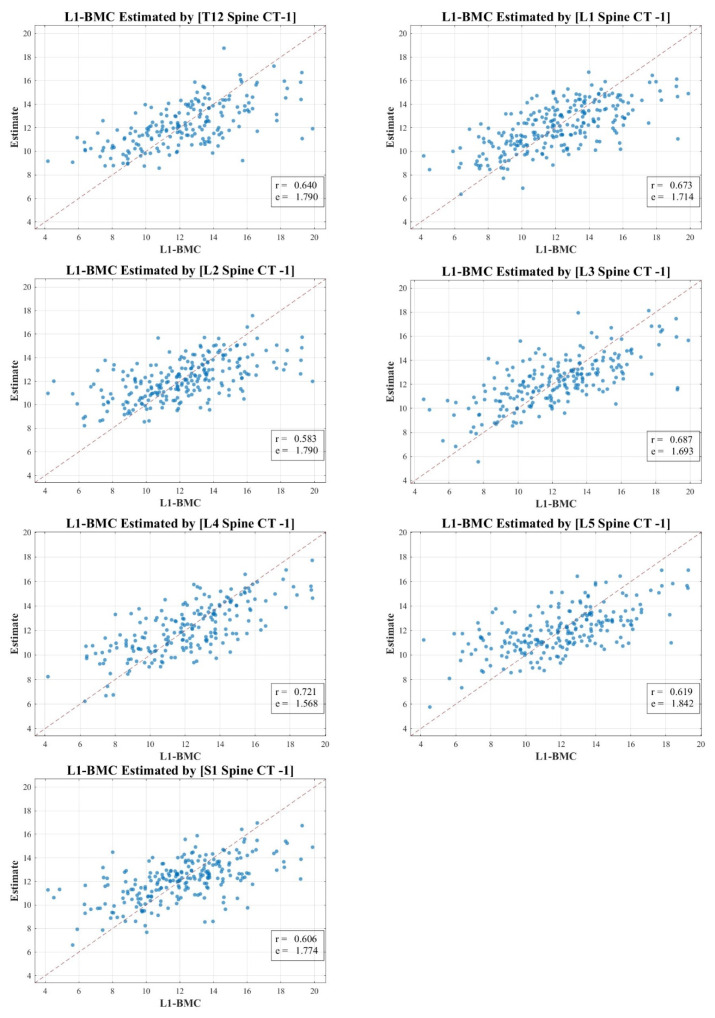
Correlation scatter plot: comparison of estimated multifidus muscle texture values from T12-S1 CT axial cuts vs. total lumbar DXA BMC.

**Figure 4 diagnostics-13-03466-f004:**
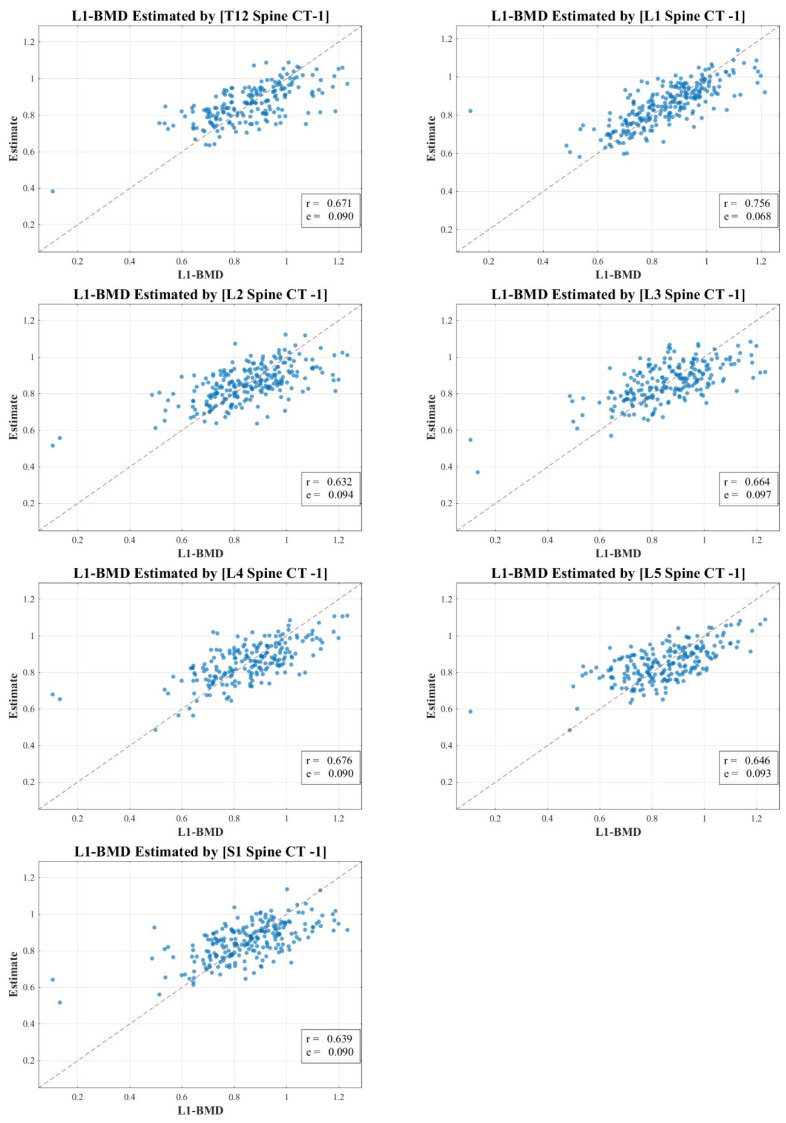
Correlation scatter plot: comparison of estimated multifidus muscle texture values from T12-S1 CT axial cuts vs. total lumbar DXA BMD.

**Figure 5 diagnostics-13-03466-f005:**
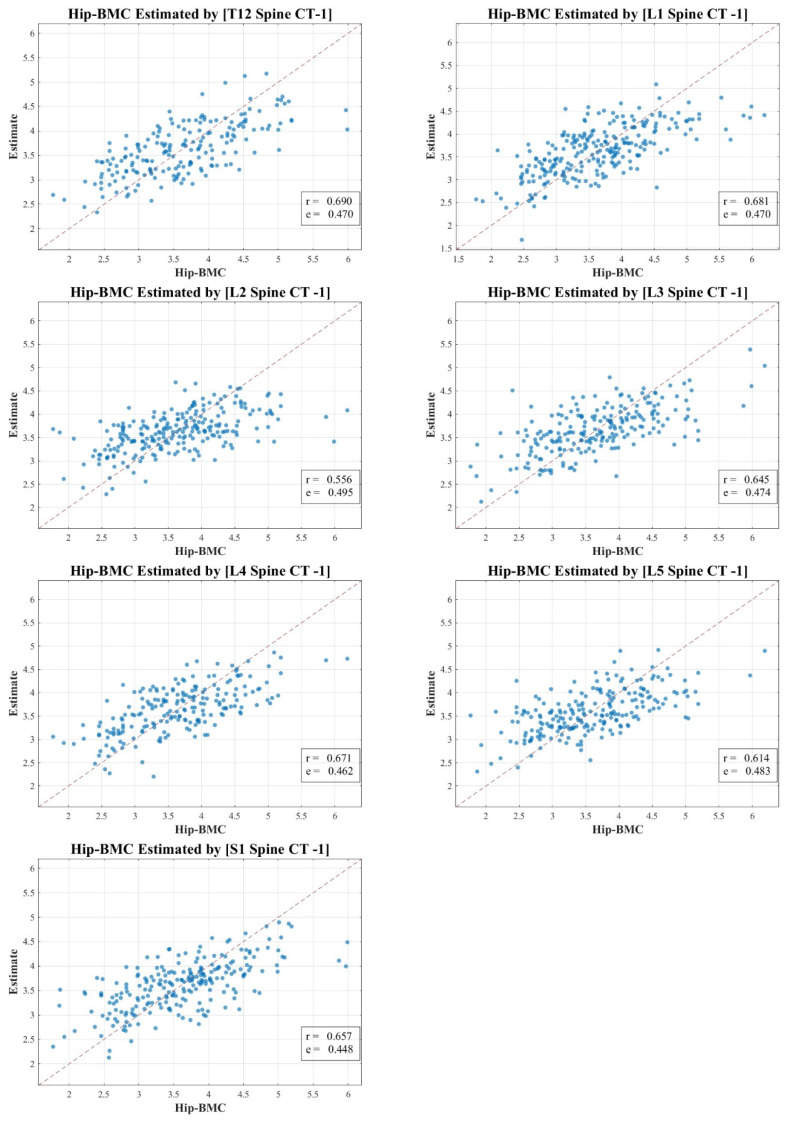
Correlation scatter plot: comparison of estimated multifidus muscle texture values from T12-S1 CT axial cuts vs. total hip DXA BMC.

**Figure 6 diagnostics-13-03466-f006:**
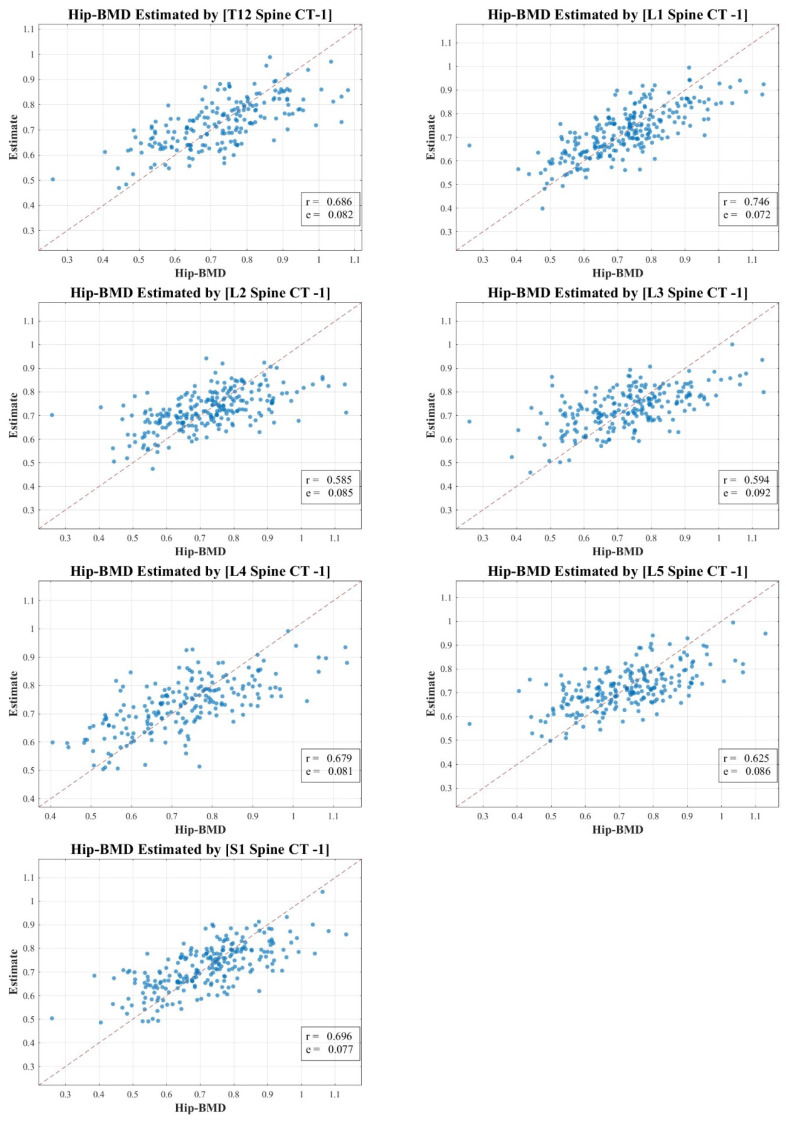
Correlation scatter plot: comparison of estimated multifidus muscle texture values from T12-S1 CT axial cuts vs. total hip DXA BMD.

**Table 1 diagnostics-13-03466-t001:** Process of patients undergoing concurrent spine CT and DXA scans.

Data Description	Number of Cases (*n*)
Assessed for eligibility	*n* = 1722 cases (863 patients)
Included in study (DXA and CT taken within a month)	*n* = 1126 (590 patients)
Spine CT (Actual measurable axial cuts for T12-S1)	*n* = 856 (417 patients)
Excluded	History of Lumbar body compression or burst fractures (*n* = 155) (45 patients) History of surgery for a previous fracture - Vertebroplasty (*n* = 46) (25 patients) - Kyphoplasty (*n* = 51) (23 patients) - Metal artifacts (*n* = 92) (31 patients) Difficulty in identifying trabecular bones due to severe osteolytic or pathological changes. (*n* = 117) (45 patients)
Final Analysis for the Study	*n* = 395 (248 patients)

**Table 2 diagnostics-13-03466-t002:** Gray-level co-occurrence matrix feature parameters.

Analytical Tool	Parameter	Value/Name/Function	Feature #
Histogram	Statistics (k)	mean (k = 1), standard deviation (k = 2), skewness (k = 3), kurtosis (k = 4) entropy (k = 5)	5
Texture (GLCM)	Directions (l)	horizontal (l = 1), vertical (l = 2)	2 × 4 × 5 = 40
	Levels (m)	16 (m = 1), 32 (m = 2), 64 (m = 3), 128 (m = 4)	
	Statistics (n)	contrast (n = 1), correlation (n = 2), energy (n = 3), homogeneity (n = 4), variance (n = 5)	

**Table 3 diagnostics-13-03466-t003:** Demographic and clinical characteristics of the study participants.

Case (Number)	395 (248)
Mean age (years)	63.12 ± 10.16
The time between CT and DXA dates (days)	7.13 ± 6.12
Sex (male/female)	115/133
BMI (kg/m^2^)	24.09 ± 4.45

## Data Availability

The datasets generated and/or analyzed during the current study are not publicly available because we did not obtain authorization from the patients for disclosure regarding patient privacy. However, datasets are available from the corresponding author on reasonable request.

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
