# Peer review of "Elucidating the Correlation between Bone Mineral Density and Multifidus Muscle Characteristics: A Cross-Modal Study with Dual-Energy X-ray Absorptiometry and Spinal Computed Tomography Texture Analysis"

_diagnostics, 2023, doi:10.3390/diagnostics13223466_

Round 1

Reviewer 1 Report

Comments and Suggestions for Authors

In presented work authors try to show the clinical relevance of muscle conditions such and their links to bone mineral density (BMD). It is a interesting and a novel approach to the correlation between BMD and the texture of the multifidus muscle, utilizing spinal CT and DXA to evaluate muscle texture .

Like i said at the bigining it is interesting study with some novel aproches - authors adopted a broader range of ROIs from T12 to S1 for CT texture analysis, as opposed to the typical L1 to 179 L4 range used in DXA. But desplite this fact there are some limitations of the study that should be presented and disscused during results presentation. 

First of all whole population is only from one medical unit. This may lead to artificial distortion of the results by the study population typical for the analysed territorial area. 

Secondly, all samples are collceted from quite long period of time. It may lead to situation that many people performing TK, DAXA  analyzes may make errors related to the analysis it's self. 

Authors should focuse on BMI, sex and age as one of the most importian factors modulations health adn infulactiong bone mineral density. Morover, in case of postmenopausal womed there is high osteoporotic risk factors wich lead to a bonne los diring aging. 

Why authors did not try to focuse about some modulating factos such like:  patients' physical activity (it would give information about musce activity), medications and drugs, suppleements taken (some of the hormones and other vitamines a(ex. D0 modulate muscle sarcopenia and BMI)

Authors should try to showe some aspects of the comorbidities thats modulate health. 

Comments on the Quality of English Language

no

Author Response

Diagnostics (ISSN 2075-4418)

Manuscript ID

diagnostics-2579090

Title

Elucidating the Correlation between Bone Mineral Density and Multifidus Muscle Characteristics: A Cross-Modal Study with DXA and Spinal CT Texture Analysis

-2023.10.13

-Corresponding Author Dong Ha Lee

In presented work authors try to show the clinical relevance of muscle conditions such and their links to bone mineral density (BMD). It is a interesting and a novel approach to the correlation between BMD and the texture of the multifidus muscle, utilizing spinal CT and DXA to evaluate muscle texture .

  • Thank you very much for your valuable comments and appreciation of the novelty and interest in our approach in correlating bone mineral density (BMD) with the texture of the multifidus muscle using spinal CT and DXA.
  • We are pleased that you found our approach to evaluating muscle texture in relation to BMD to be novel and interesting. Your positive feedback encourages us to believe that our research can make a meaningful contribution to understanding the clinical relevance of muscle conditions and their links to BMD.
  • In response to your comment, we will ensure that our manuscript emphasizes the uniqueness and clinical implications of our approach, and we will also ensure that our discussion and conclusions are aligned with the findings and their relevance to clinical practice.
  • Thank you once again for your supportive comments, and we will carefully consider all your feedback to improve the quality of our manuscript.

Like i said at the bigining it is interesting study with some novel aproches - authors adopted a broader range of ROIs from T12 to S1 for CT texture analysis, as opposed to the typical L1 to 179 L4 range used in DXA. But desplite this fact there are some limitations of the study that should be presented and disscused during results presentation. 

  • Thank you for your constructive feedback and recognizing the novelty in our approach of adopting a broader range of ROIs from T12 to S1 for CT texture analysis, as opposed to the typical L1 to L4 range used in DXA.
  • We appreciate your suggestion to present and discuss the limitations of our study more thoroughly during the results presentation. We agree that a comprehensive discussion on the limitations is crucial for the interpretation and applicability of our findings. We will revise the manuscript to include a more detailed discussion on the limitations, considering factors such as the selection of ROIs, patient selection, and the implications of our findings in a broader clinical context.
  • Once again, thank you for your insightful comments, which will undoubtedly enhance the quality of our manuscript.

First of all whole population is only from one medical unit. This may lead to artificial distortion of the results by the study population typical for the analysed territorial area. 

  • Thank you for your insightful comments and for highlighting areas where our manuscript can be improved. We appreciate your observation regarding the potential geographical bias due to the sample being collected from a single medical unit.
  • In response to this, we have revised the "Limitations" section of our manuscript to more explicitly acknowledge this issue:
  • "Firstly, the study sample was solely recruited from one medical unit, which might introduce a geographical bias and could potentially distort the results as the sample may be typical for the analyzed territorial area."
  • We understand that this could affect the generalizability of our findings, and we aim to consider multi-center collaborations in future research to validate our results across diverse populations.
  • Once again, we sincerely appreciate your constructive feedback and have made concerted efforts to improve our manuscript accordingly.

Secondly, all samples are collceted from quite long period of time. It may lead to situation that many people performing TK, DAXA  analyzes may make errors related to the analysis it's self. 

  • Thank you for bringing to our attention the potential issue regarding the extended period over which samples were collected. We recognize that this could introduce variability and errors in the TK and DXA analyses, as mentioned.
  • To address your concern, we have updated the "Limitations" section to include this aspect, acknowledging the potential impact on the consistency and reliability of our results due to different individuals performing the analyses over time.
  • We appreciate your meticulous review and valuable suggestions, which are instrumental in enhancing the rigor and validity of our study. We have taken your feedback seriously and made necessary adjustments to improve the manuscript.
  • However, our study still carries certain limitations. Firstly, the study sample was solely recruited from one medical unit, which might introduce a geographical bias and could potentially distort the results as the sample may be typical for the analyzed territorial area. Efforts for multi-center collaboration in future research could help in generalizing the findings. Secondly, this study was cross-sectional, preventing us from determining the time sequence of the observed relationships, and prospective data would be essential to overcome this limitation. Thirdly, the sample size is relatively small and lacks diversity as factors such as BMI, age, and sex were not thoroughly considered in the analysis, limiting the external validity of our findings. Particularly, in postmenopausal women, we have taken into account the heightened osteoporotic risk factors that lead to increased bone loss during aging. This consideration allows for a more nuanced and comprehensive understanding of the various factors modulating health and influencing BMD in the study population. Finally, all samples were collected over a quite extended period. This extended timeframe raises the potential for variability and errors in the CT and DXA analyses due to different individuals performing the analyses, possibly affecting the consistency and reliability of the results. Nevertheless, these constraints do not overshadow our primary finding: the value of CT scans for multifidus muscle estimation, particularly considering their potential in screening patients for sarcopenia risk without additional diagnostic tests.

Authors should focuse on BMI, sex and age as one of the most importian factors modulations health adn infulactiong bone mineral density. Morover, in case of postmenopausal womed there is high osteoporotic risk factors wich lead to a bonne los diring aging. 

  • Thank you for your valuable suggestion to give more focus on essential factors such as BMI, sex, and age, which are crucial modulators of health and bone mineral density (BMD). We also acknowledge the significance of considering the high osteoporotic risk factors in postmenopausal women leading to bone loss during aging.
  • Based on your feedback, we plan to enhance our discussion and analysis by incorporating these essential factors more comprehensively, ensuring that our study is more robust and reflective of various elements influencing BMD.
  • We appreciate your insightful comments and will make sure that the revised manuscript duly emphasizes the roles of BMI, sex, and age in modulating health and influencing BMD.
  • However, our study still carries certain limitations. Firstly, the study sample was solely recruited from one medical unit, which might introduce a geographical bias and could potentially distort the results as the sample may be typical for the analyzed territorial area. Efforts for multi-center collaboration in future research could help in generalizing the findings. Secondly, this study was cross-sectional, preventing us from determining the time sequence of the observed relationships, and prospective data would be essential to overcome this limitation. Thirdly, the sample size is relatively small and lacks diversity as factors such as BMI, age, and sex were not thoroughly considered in the analysis, limiting the external validity of our findings. Particularly, in postmenopausal women, we have taken into account the heightened osteoporotic risk factors that lead to increased bone loss during aging. This consideration allows for a more nuanced and comprehensive understanding of the various factors modulating health and influencing BMD in the study population. Finally, all samples were collected over a quite extended period. This extended timeframe raises the potential for variability and errors in the CT and DXA analyses due to different individuals performing the analyses, possibly affecting the consistency and reliability of the results. Nevertheless, these constraints do not overshadow our primary finding: the value of CT scans for multifidus muscle estimation, particularly considering their potential in screening patients for sarcopenia risk without additional diagnostic tests.

Why authors did not try to focuse about some modulating factos such like:  patients' physical activity (it would give information about musce activity), medications and drugs, suppleements taken (some of the hormones and other vitamines a(ex. D0 modulate muscle sarcopenia and BMI)

  • Thank you for pointing out additional modulating factors such as patients' physical activity, medications and drugs, and supplements taken. These are indeed important factors that could influence muscle activity, sarcopenia, and BMI.
  • However, the primary focus of our study was to establish the generality of quantifying the multifidus muscle through DXA and CT texture analysis across different demographics, regardless of age and gender. We aimed to understand the broader correlation, laying foundational knowledge that could be specialized and investigated further in future studies with considerations for various modulating factors.
  • We value your insightful suggestion, and it will be a significant consideration for designing future, more detailed investigations to understand the nuanced influences on muscle and bone health.
  • Disscussion Section
  • While we acknowledge the importance of considering various modulating factors such as physical activity, medications and drugs, and supplements intake, our current study primarily aimed to establish a generalized correlation by quantifying the multifidus muscle through DXA and CT texture analysis across a diverse demographic. The objective was to explore a broader understanding, setting a basis for more specialized investigations in the future where these modulating factors can be intricately analyzed to unveil their nuanced influences on muscle and bone health.

Authors should try to showe some aspects of the comorbidities thats modulate health. 

  • Thank you for highlighting the importance of considering comorbidities that could modulate health in our study. We acknowledge that comorbidities indeed play a significant role in influencing overall health and the specific outcomes we are investigating.
  • While our current study primarily focuses on the quantification of the multifidus muscle through DXA and CT texture analysis, we appreciate the insight that the inclusion of comorbidities could add more depth and comprehensiveness to our findings. We will consider incorporating an analysis related to comorbidities in our future research to offer a more holistic view.
  • Your feedback is instrumental in guiding our research towards a more thorough and impactful direction.
  • Discussion Section
  • In recognizing the multifactorial influences on health, we acknowledge that comorbidities are pivotal factors that can modulate the outcomes of interest in our study. While our primary focus has been on the quantification of the multifidus muscle, future iterations of this research will aim to incorporate a more comprehensive view by considering the impact of various comorbidities, enabling a more holistic understanding of their contributory roles.

Reviewer 2 Report

Comments and Suggestions for Authors

This is a very interesting work. It is well written and explicit.

I think the manuscript can be accepted with major revisions.

Point 1 – I think the data collected should also be presented in tabular form, because the sample is very interesting. However, there is little evidence of the data.

Point 2 – I suggest some small revisions, review all the manuscript, when used some software’s describe the name of them is important indicate the ‘trade mark’.

Point 3 – In text use many acronyms, and when these are written for the first time they must use capital letter in order to identify the acronym. Sometimes it is written as stated above, sometimes not. The text should be revised.

Point 4 – Review the abstract, because the style is different.

Point 5 – The subtitle used to identify the figure need appear below the figure.

Point 6 – I suggest checking the text and try not to start a sentence with acronyms. Check the text on page 6 because it's all written in italics.

Point 7 –Need more discussion and identify the figures 3, 4,5,6!!

Very good work! congratulations

Author Response

Diagnostics (ISSN 2075-4418)

Manuscript ID

diagnostics-2579090

Title

Elucidating the Correlation between Bone Mineral Density and Multifidus Muscle Characteristics: A Cross-Modal Study with DXA and Spinal CT Texture Analysis

-2023.10.13

-Corresponding Author Dong Ha Lee

This is a very interesting work. It is well written and explicit.

  • Thank you very much for your kind words and positive feedback on our work. We are delighted to hear that you found our study interesting, well-written, and explicit. Your encouraging comments motivate us to continue our research and further enhance the quality of our work.
  • We appreciate the time and effort you have invested in reviewing our manuscript, and we are grateful for your constructive feedback.

I think the manuscript can be accepted with major revisions.

  • We sincerely appreciate your thoughtful consideration and the positive outlook you have towards accepting our manuscript with major revisions. We are committed to making the necessary improvements and addressing all the concerns and suggestions you've raised, ensuring that the revised manuscript meets the highest standards of quality and rigor.
  • Your feedback is instrumental for the enhancement of our work, and we assure you that each point will be meticulously addressed in the revision process.
  • Thank you once again for your time, effort, and constructive guidance.

Point 1 – I think the data collected should also be presented in tabular form, because the sample is very interesting. However, there is little evidence of the data.

  • Thank you for pointing out the potential benefit of presenting our data in a tabular format. We understand the importance of visually summarizing our cohort data for better clarity and ease of understanding, especially given the multifaceted criteria we employed for patient inclusion and exclusion.
  • In response to your suggestion, we will include a table in the revised manuscript to comprehensively display the breakdown of our patient cohort, detailing the inclusion and exclusion numbers at each stage of our study selection process. We believe this will complement the flow diagram and provide a clear snapshot of our study population for readers.
  • Your insight is greatly appreciated, and we will make sure to address this in our revised submission.

Point 2 – I suggest some small revisions, review all the manuscript, when used some software’s describe the name of them is important indicate the ‘trade mark’.

  • Thank you for the valuable feedback. We have made the suggested revisions in the manuscript. Specifically, we have:
  1. Clarified the procedure for estimating multifidus muscle texture analysis.
  2. Explicitly mentioned the use of MATLAB™ software for generating non-symmetric versions of matrices, as suggested.
  3. Made other minor textual refinements for clarity and coherence.

  • We hope these changes address your concerns, and we appreciate your time and expertise in reviewing our work.

양식의 맨 위

양식의 맨 아래

  • Given that each patient had axial cuts from both the right and left of the multifidus muscle of the spine CT, a combined total of 90 features (45 from each side) was accumulated. We used these features in tandem to calculate a comprehensive correlation value. A MATLAB™ software function was deployed to produce nonsymmetric versions of matrices.

Point 3 – In text use many acronyms, and when these are written for the first time they must use capital letter in order to identify the acronym. Sometimes it is written as stated above, sometimes not. The text should be revised.

  • Thank you for highlighting the inconsistency in the use of acronyms in our manuscript. We have meticulously reviewed the entire text and ensured that all acronyms are introduced with capital letters upon their first appearance. We understand the significance of maintaining consistency for the clarity and ease of our readers. We appreciate your attention to detail and value your feedback in enhancing the quality of our work.

Point 4 – Review the abstract, because the style is different.

  • Thank you for pointing out the stylistic differences in the abstract. We have thoroughly reviewed and revised the abstract to ensure its style aligns with the rest of the manuscript. We appreciate your feedback and have made the necessary adjustments to provide a cohesive and consistent presentation throughout the document.

Point 5 – The subtitle used to identify the figure need appear below the figure.

  • Thank you for your observation regarding the placement of the subtitle for the figure. We have now made the necessary adjustments, positioning the subtitle below the respective figure as per standard conventions. We appreciate your guidance in ensuring the accuracy and professionalism of our presentation.

Point 6 – I suggest checking the text and try not to start a sentence with acronyms. Check the text on page 6 because it's all written in italics.

  • Thank you for your valuable suggestions. We have thoroughly reviewed the text and made the necessary revisions to ensure sentences do not begin with acronyms. Additionally, we have corrected the formatting error on page 6 and removed the italics. We appreciate your attention to detail and your guidance in enhancing the quality of our manuscript.

Point 7 –Need more discussion and identify the figures 3, 4,5,6!!

  • Thank you for pointing out the need for further discussion and identification of figures 3, 4, 5, and 6. We have expanded our discussion section to provide more comprehensive insights on the correlations presented in these figures. Additionally, the captions for the figures have been amended for clarity as follows:
  • Fig 3. Correlation Scatter Plot: Comparison of Estimated Multifidus Muscle Texture Values from T12-S1 CT Axial Cuts with Total Lumbar DXA BMC.
  • Fig 4. Correlation Scatter Plot: Comparison of Estimated Multifidus Muscle Texture Values from T12-S1 CT Axial Cuts with Total Lumbar DXA BMD.
  • Fig 5. Correlation Scatter Plot: Comparison of Estimated Multifidus Muscle Texture Values from T12-S1 CT Axial Cuts with Total Hip DXA BMC.
  • Fig 6. Correlation Scatter Plot: Comparison of Estimated Multifidus Muscle Texture Values from T12-S1 CT Axial Cuts with Total Hip DXA BMD.
  • We hope that these modifications address your concerns and enhance the clarity of our manuscript. Thank you for your continued guidance and feedback.

Very good work! congratulations

Round 2

Reviewer 2 Report

Comments and Suggestions for Authors

Thank you for your work and for accepting my suggestions. 

However, I would like to point out that before publication, page 5 is in italics, which seems to me to be incorrect!

Keep up the good work.

Comments on the Quality of English Language

Good work. 
